# Prognosis of Hypothermic Patients Undergoing ECLS Rewarming—Do Alterations in Biochemical Parameters Matter?

**DOI:** 10.3390/ijerph18189764

**Published:** 2021-09-16

**Authors:** Hubert Hymczak, Paweł Podsiadło, Sylweriusz Kosiński, Mathieu Pasquier, Konrad Mendrala, Damian Hudziak, Radosław Gocoł, Dariusz Plicner, Tomasz Darocha

**Affiliations:** 1Department of Anesthesiology and Intensive Care, John Paul II Hospital, 31-202 Krakow, Poland; hymczak@op.pl; 2Faculty of Medicine and Health Sciences, Andrzej Frycz Modrzewski Krakow University, 30-705 Krakow, Poland; 3Institute of Medical Sciences, Jan Kochanowski University, 25-369 Kielce, Poland; p.podsiadlo.01@gmail.com; 4Faculty of Health Sciences, Jagiellonian University Medical College, 31-008 Krakow, Poland; kosa@mp.pl; 5Emergency Department, Lausanne University Hospital, University of Lausanne, 1015 Lausanne, Switzerland; mathieu.pasquier@chuv.ch; 6Department of Anaesthesiology and Intensive Care, Medical University of Silesia, 40-055 Katowice, Poland; k.mendrala@gmail.com (K.M.); tomekdarocha@wp.pl (T.D.); 7Department of Cardiac Surgery, Medical University of Silesia, 40-055 Katowice, Poland; damhud@gmail.com (D.H.); gocot@poczta.onet.pl (R.G.); 8Unit of Experimental Cardiology and Cardiac Surgery, Faculty of Medicine and Health Sciences, Andrzej Frycz Modrzewski Krakow University, 30-705 Krakow, Poland

**Keywords:** ECLS, accidental hypothermia, lactate, lactate kinetics, rewarming

## Abstract

Background: While ECLS is a highly invasive procedure, the identification of patients with a potentially good prognosis is of high importance. The aim of this study was to analyse changes in the acid-base balance parameters and lactate kinetics during the early stages of ECLS rewarming to determine predictors of clinical outcome. Methods: This single-centre retrospective study was conducted at the Severe Hypothermia Treatment Centre at John Paul II Hospital in Krakow, Poland. Patients ≥18 years old who had a core temperature (Tc) < 30 °C and were rewarmed with ECLS between December 2013 and August 2018 were included. Acid-base balance parameters were measured at ECLS implantation, at Tc 30 °C, and at 2 and 4 h after Tc 30 °C. The alteration in blood lactate kinetics was calculated as the percent change in serum lactate concentration relative to the baseline. Results: We included 50 patients, of which 36 (72%) were in cardiac arrest. The mean age was 56 ± 15 years old, and the mean Tc was 24.5 ± 12.6 °C. Twenty-one patients (42%) died. Lactate concentrations in the survivors group were significantly lower than in the non-survivors at all time points. In the survivors group, the mean lactate concentration decreased −2.42 ± 4.49 mmol/L from time of ECLS implantation until 4 h after reaching Tc 30 °C, while in the non-survivors’ group (*p* = 0.024), it increased 1.44 ± 6.41 mmol/L. Conclusions: Our results indicate that high lactate concentration is associated with a poor prognosis for hypothermic patients undergoing ECLS rewarming. A decreased value of lactate kinetics at 4 h after reaching 30 °C is also associated with a poor prognosis.

## 1. Introduction

Accidental hypothermia is a specific cause of cardiac arrest (CA) of which a survival rate in witnessed CA is about 56% and is approximately 27% in unwitnessed CA, mostly with good neurologic outcome [1,2]. Currently, extracorporeal life support (ECLS) rewarming is the method of choice in the treatment of out-of-hospital cardiac arrest patients (OHCA) with severe accidental hypothermia [3]. However, ECLS is a highly invasive procedure, fraught with the risk of serious complications and is not broadly accessible [4]. Therefore, the identification of patients with a potentially good prognosis is of high importance. For years, the serum potassium concentration has been used to indicate patients who have potential for survival [5]. Recently, the “HOPE score” was developed to predict the outcome among OHCA patients with hypothermia if ECLS rewarming is applied [1]. However, this calculator includes data from the prehospital period, and its main value is to help clinicians decide whether to continue resuscitation efforts with ECLS rewarming. Apart from CA patients, patients with cardiac instability caused by severe accidental hypothermia can also be successfully rewarmed with ECLS [3]. To the best of our knowledge, no criteria have yet been developed to estimate the chances of survival of severely hypothermic patients, with or without cardiac arrest, after the initialization of extracorporeal treatment. It would be of great value to find means to identify the high-risk patients earlier in whom the therapy should be intensified or modified. Recent studies by Darocha et al. and Podsiadlo et al. highlight the important role of the acid-base balance parameters and the lactate concentration in the prognosis of victims of accidental hypothermia rewarmed with ECLS [6,7]. Therefore, it seems likely that observing the changes in these parameters over time can help predict the patients’ outcome. The aim of this study was to analyse the changes in the acid-base balance parameters and lactate kinetics during the early stages of ECLS rewarming to indicate the predictors of clinical outcome.

## 2. Materials and Methods

This single-centre retrospective study was conducted at the Severe Hypothermia Treatment Centre (SHTC) at John Paul II Hospital in Krakow, Poland. Data were collected from available medical records in paper and electronic form. The primary outcome was the survival to discharge from the intensive care unit. The individual variables and treatment outcomes were analysed by the comparison of the surviving patient group to the patients that died. Study approval was obtained from the Bioethics Committee of the Jagiellonian University Medical College, Krakow, Poland (no. 1072.6120.344.2018).

### 2.1. Inclusion and Exclusion Criteria

Patients with accidental hypothermia selected for this study were over the age of 18 and referred to SHTC by telephone, by emergency medical services (EMS), as well as by the hospital emergency department team between December 2013 and August 2018. Based on their medical history, only patients subsequently qualified for extracorporeal therapy were included in this study.

The exclusion criteria were as follows: core temperature (Tc) ≥ 30 °C, veno-arterial Extra Corporeal Membrane Oxygenation (v-a ECMO) implantation outside SHTC, disqualification from v-a ECMO therapy, and initiation of renal replacement therapy less than 12 h after v-a ECMO implantation.

Severe hypothermia was confirmed by measuring the oesophageal temperature at hospital admission. Haemodynamic instability was diagnosed on the basis of at least one of the following criteria: systolic blood pressure <90 mmHg and/or ventricular arrhythmia.

### 2.2. Data Collection

The following patient information was collected: age, gender, Tc, heart rate, blood pressure, and in case of sudden CA, the time of resuscitation to initiation of v-a ECMO; circumstances of hypothermia, rate of rewarming, duration of ECLS, duration of mechanical ventilation, fluid load and diuresis in the first 24 h, use of catecholamines, duration of Intensive Care Unit (ICU) stay, and cerebral performance category scale (CPC), which was assessed at discharge from the ICU.

Biochemical parameters were measured at the following time points: T1—at ECMO implantation; T2—obtained Tc 30 °C; T3—2 h after Tc 30 °C; and T4—4 h after Tc 30 °C. Alpha-stat arterial blood gas analysis (pH, PaCO_2_, PaO_2_, and concentrations of HCO_3_, BE, potassium, haemoglobin, glucose, and lactate) was performed at all time points. The value of blood lactate kinetics was calculated as the percent change in serum lactate concentration at time points T2, T3, and T4 relative to the baseline lactate concentration at time point T1 and to the relative preceding time point, according to the following formula [8]:(1)$$\mathrm{Lactate}\;\mathrm{kinetics}\;\mathrm{at}\;\mathrm{time}\;\mathrm{point}\;(\%)=\frac{initial\;LAC-subsequent\;LAC}{initial\;LAC\times 100}$$

A positive value of lactate kinetics represents a decrease in serum lactate concentration, while a negative lactate kinetics indicates an increase in serum lactate concentration at consecutive time points. Tests were performed in a Randox International Quality Assessment Scheme (RIQAS) certified laboratory. Samples were routinely analysed with the Cobas 6000 (Roche Diagnostic GmbH, Manheim, Germany) and the ABL835 FLEX (Radiometer Medical ApS, Brønshøj, Copenhagen, Denmark) analysers.

### 2.3. Clinical Procedure

In all cases, v-a ECMO therapy was initially performed using a Rotaflow Console REF 706037 (Maquet, Rastatt, Germany) with a Medos Deltastream HC heat exchanger (Inspiration Healthcare Group, Cravley, UK) and then continued with the heat unit HU 35 (Maquet, Rastatt, Germany) and a permanent life support set oxygenator (Maquet, Rastatt, Germany) that used an HLS Cannulae with BIOLINE Coating (Maquet, Rastatt, Germany). Implantation via peripheral access was performed through insertion of a 17–21 F size arterial cannula in the right or left femoral artery and a 22–24 F size venous cannula through the right or left femoral vein. The rewarming rate was individually determined according to the patients’ clinical needs. Patients were transferred to the ICU once their heart rate stabilised and a body temperature of ≥32 °C was achieved.

V-a ECMO support was removed when normothermia was achieved and the circulatory system was stabilised after at least six hours of therapy. In cases of lower limb ischaemic complications or massive bleeding at the implantation site, the support was removed before six hours. If normothermia was achieved with haemodynamic instability, persistent asystole, and inability to maintain v-a ECMO output, death was pronounced. Catecholamines (epinephrine, norepinephrine, dobutamine) were needed for all patients. Red blood cell concentrate was transfused to achieve a haemoglobin concentration ≥10 g/dL as required.

### 2.4. Statistical Analysis

We compared the values of biochemical parameters between the survivors and non-survivors groups at each time point. We also calculated the differences in these values between consecutive time points in each group and then compared these trends between groups. The quantitative variables are presented as the mean with standard deviation (SD) or the median (M) and the lower (q1) and upper (q3) quartiles depending on the distribution of the variable. Normal distribution was investigated using the Shapiro–Wilk test. The distribution of qualitative variables is described as the absolute and relative frequencies. The relationship between two qualitative features was analysed with the chi-square test. Significance of the quantitative traits between the two groups was analysed using Student’s *t*-test (the distribution follows the normal distribution) or Mann–Whitney U test (the distribution deviates from the normal distribution). The repeated-measures ANOVA test was used to compare the trends of biochemical parameters at consecutive time points between the groups. Multivariate logistic regression was then used to identify predictors of survival. The results are presented as odds ratio (OR) and a 95% confidence interval (95% CI). Additionally, a moderation analysis was performed to investigate if CA occurrence influenced the relationship between survival and the studied parameters. Since no significant interaction was observed, patients with preserved circulation and those in cardiac arrest were not analysed separately. Analyses were performed using IBM SPSS Statistics for Windows, version 25 (IBM Corp., Armonk, NY, USA) and Statistica 12.0 (Statsoft, Tulsa, OK, USA). A level of α = 0.05 for two-tailed tests was used to indicate significance.

## 3. Results

Fifty-two patients met the eligibility criteria. Two patients were excluded from the study because the v-a ECMO was not implanted at the SHTC, and a different treatment regimen was used in conjunction with incomplete medical and laboratory data. The remaining 50 patients (mean age of 56 ± 14.6 years) were included. The youngest patient was 19 years old and the oldest was 84. Thirty-nine of the patients were men (78%). The mean Tc on admission was 24.5 °C (SD 2.6; range 16.9–29 °C). This study included two drowning victims (4%) and one patient (2%) buried in an avalanche (extricated with preserved circulation). The remaining patients suffered from exposure to cold air. However, specific data regarding these events and the detailed patient characteristics are incomplete and not suitable for analysis. Thirty-six patients (72%) were in CA at the time of cannulation, and fourteen (28%) had preserved circulation. Among the patients who experienced CA, in 24 cases (66.7%) the initial heart rhythm was ventricular fibrillation (VF), in 11 cases asystole (30.5%), and pulseless electrical activity (PEA) occurred in one case (2.8%). In seven cases (19.4%) cardiac arrest occurred before the victim was found. The median resuscitation time was 140 min (IQR 102–202.5). The mean systolic pressure among patients with preserved circulation was 67.14 ± 10.69 mmHg. The median duration of v-a ECMO therapy was 21.5 h (IQR 8.33–33). The median rewarming rate was 1.8 °C/h (range: 0.4–6 °C/h).

Twenty-one study participants (42%) died, and of those, four died before the T4 time point. Men accounted for 71% of the deceased patients. Twenty-seven patients who survived (93%) were scored with CPC-1 and two (7%) with CPC-2. The measured parameters at admission between the survivors and those that died are summarised in Table 1.

Numerical values are presented in the following order: the number of patients or the mean or median values and the percentage (%) or standard deviation (±) or lower and upper quartile (q1–q3). ICU—Intensive Care Unit, Tc—core temperature, CA—cardiac arrest, v-a ECMO—veno-arterial Extra Corporeal Membrane Oxygenation.

### 3.1. Changes of Concentrations in the Biochemical Parameters

The blood pH, PaCO_2_, PaO_2_, HCO_3_, BE, and Hgb values changed significantly between subsequent time points in each group, although there were no significant differences in these values between the survivors and non-survivors groups (Figure 1).

The survivors group blood pH was significantly higher than that of the non-survivors group at time point T4 (7.23 ± 0.16 vs. 7.12 ± 0.17; *p* = 0.037). There was no statistically significant difference in PaO_2_ and PaCO_2_ values between the groups at any time point. A significantly higher HCO_3_ concentration was observed in the survivors group at time point T4 (12.70 ± 4.42 vs. 9.80 ± 4.07; *p* = 0.03). The survivors also had a significantly higher BE than non-survivors at time point T4 (−13.60 ± 6.55 vs. −18.20 ± 6.69; *p* = 0.029). The serum potassium concentrations were significantly lower in the survivors group at all time points. However, these concentrations did not change significantly in the consecutive time points.

The lactate concentrations in the survivors group were significantly lower at all time points than in the non-survivors. There was also a statistically significant difference in the lactate decrease between time points T4 and T1 in the survivors group compared to the non-survivors. A significant decrease in the mean lactate concentrations occurred among the survivors, while no significant change was observed in the non-survivors. A significant difference between groups in lactate kinetics was observed for T4-T1 and T4-T3 values. The difference in lactate kinetics value for the other time points did not reach statistical significance (Table 2). The highest lactate concentration registered in a survivor was 21 mmol/L, and which was measured at the T1 time point in a patient with preserved spontaneous circulation.

A comparison of clinical parameters between survivors and those who died is shown in Table 3. All patients required catecholamine supply early on during v-a ECMO therapy. The rewarming rate ranged from 0.4–6 °C/h and did not differ significantly between groups.

Lactate concentration is expressed in (mmol/L) as a mean and standard deviation (±). Lactate kinetics is expressed in % as median and lower and upper quartile (q1–q3). T1—time at ECMO implantation, T2—time obtained Tc 30 °C, T3—2 h after Tc 30 °C, and T4—4 h after Tc 30 °C.

Numerical values are presented in the following order: median, lower and upper quartiles (q1–q3) or mean value, standard deviation or numerical value, percentage. NaHCO3—sodium bicarbonate, RCC—Red cell concentrate, v- a ECMO—veno-arterial Extra Corporeal Membrane Oxygenation, ICU—Intensive Care Unit.

### 3.2. Predictors of Survival

The strongest predictor associated with survival was observed for lactate concentration at T4 point. This was determined with a receiver operating characteristic curve (ROC) analysis for the occurrence of death in relation to the lactate concentration at time point T4 and is presented in Figure 2. We found that a lactate concentration above 12.55 mmol/L at the T4 time point increases the risk of death.

In the survival analysis with logistic regression, the model with the highest predictive value included: age, gender, CA occurrence, Tc, rewarming rate, and lactate concentration at the analysed time points (Table 4). In this set of data, the patients’ gender and lactate concentrations appeared to be independently associated with survival. The area under the ROC curve of this model was 0.911 (95% CI 0.829–0.993). A 1-mmol/L higher lactate concentration at the T1 time point was associated with a 15% reduced chance of survival (OR = 0.85, 95% CI = (0.735–0.986)). In contrast, a 1-mmol/L higher lactate concentration at the T4 time point was associated with a 36% reduced chance of survival (OR = 0.640, 95% CI 0.468–0.875). A similar predictive value was found for the model considering differences in lactate concentrations at T4 and T1 instead of absolute serum concentration (AUC = 0.906).

Data presented as odds ratio (OR), 95% confidence interval (95% CI), the area under the curve (AUC). T1—time at ECMO implantation, T2—time obtained Tc 30 °C, T3—2 h after Tc 30 °C, and T4—4 h after Tc 30 °C. CA—cardiac arrest. T_c_—core temperature.

## 4. Discussion

The results of the study showed that the lactate concentration remained a significant predictor of survival at each time point analysed. The highest lactate concentration in a surviving patient was 21 mmol/L. A significant difference between groups in the lactate kinetics value was observed for T4 versus T1 (within 4 h of achieving Tc 30 °C) and T4 versus T3 (within the fourth and second hours). The difference in lactate kinetics for all other time points failed to achieve statistical significance. Based on changes in serum lactate concentrations, we were able to calculate the chance of survival in hypothermic patients undergoing ECPR. The highest prognostic value was demonstrated for a lactate level within 4 h of achieving Tc 30 °C and for the difference with the baseline value.

The evidence that lactate is a marker of illness severity in most situations of physiological stress is compelling. Hyperlactatemia is a characteristic feature of all shock states and the level of increase in lactate concentrations is directly related to the severity of shock and to mortality [9,10,11]. Lower initial serum lactate and higher pH were associated with a better neurologic outcome in survivors of hypothermic OHCA treated with ECPR [2,12,13]. A dynamic evaluation of serial lactate concentrations may be more informative than a single value [14,15]. A better prognosis was observed with decreasing lactate concentrations in most situations of hyperlactatemia in heterogeneous patient populations [11].

It should be noted that a significant number of publications describe lactate kinetics by the term “clearance”; however, as pointed out by Vincent et al., this may be misleading [11]. Plasma lactate concentration depends on changes in production and elimination, while the term clearance refers only to the elimination of a substance from the plasma per unit time. It is therefore suggested that the phrase lactate kinetics be used instead of lactate clearance. In this regard, the existing publications can be analysed with a better understanding.

The lactate clearance (more accurately, the value of lactate kinetics) rate has been suggested as a guide for clinicians in their decision to discontinue ECPR [16,17]. It therefore seems that following the lactate trend in patients may have a prognostic value and might be an indicator of ECPR’s effectiveness. In a recent study, lactate clearance calculated through arterial blood gas analysis 6 h after ECPR proved to be one of the most important predictors of in-hospital mortality in patients treated with ECPR after cardiac arrest [15].

The effect of hyperlactatemia is governed by its severity and the clinical context. Mortality is increased by a factor of nearly three when lactic acidosis accompanies low-flow states or sepsis, and the higher the lactate level, the worse the outcome [9,10,18]. Although hyperlactatemia is often attributed to tissue hypoxia, it can result from other mechanisms including increased glycolysis, catecholamine-stimulated Na–K pump activity, alterations in pyruvate dehydrogenase activity, and reduced lactate clearance, primarily because of liver hypoperfusion [11,19].

Hypothermia-related hyperlactatemia has been observed at least since the 1980s [20]. A longer stay in a hospital of patients with lactic acidosis indicates a more severe course of hypothermia [21,22]. In a retrospective analysis from the French Alps, the median of lactate in non-survivors of hypothermic CA was almost double that of survivors [23]. Among adult OHCA patients with moderate-to-severe hypothermia, the predictive accuracy of pH of 6.9 and lactate of 13 mmol/L for 1-month survival can be useful cutoff points to rule out 1-month survival with high sensitivity [24].

The complexity of lactate metabolism during physiological stress is overwhelming, however, its complex nature makes it difficult to define what goal it should be a marker or target of [19]. In our study, we showed that a decrease in serum lactate concentration during treatment (i.e., a positive value of blood lactate kinetics) is associated with a better prognosis. However, interpreting this effect is difficult. Based on the assumption that hypothermia-related hyperlactatemia results from hypoperfusion, optimizing systemic blood flow by means of ECPR could therefore revert ongoing hypoperfusion and improve prognosis. However, a number of elements confound the clinical use of lactate and the value of blood lactate kinetics. The best known in clinical practice is the use of catecholamines, alkalizing medications, lactate-buffered CRRT, liver dysfunction, and lung lactate production. In addition, the use of specific drugs and toxins has been associated with increased lactate levels (metformin, ethanol, steroids) [19,25].

In our study, we defined the time point T2 as the moment of reaching the core temperature of 30 °C, as it is considered the potential threshold of cardiovascular stabilization and the point for unaltered catecholamines action. According to the resuscitation guidelines, a core temperature of 30 is the moment of pharmacotherapy initiation, increasing the chances of successful defibrillation and return of spontaneous circulation [26,27,28]. These assumptions allowed us to standardize the study population in our study regarding the core temperature and the stage of resuscitation. We limited our study to 4 h after the time point T2 because the number of deaths significantly reduced the group, which may have affected the statistical analysis.

Our findings indicate that based on lactate concentration and the value of lactate kinetics, it is possible to identify those patients whose prognosis is worse. A positive value of lactate kinetics may also indicate the coexistence of other factors in addition to hypoperfusion that increase lactate levels. However, considering the current state of knowledge, seeking to lower lactate levels or modifying their kinetics has no credibility [19]. The presented study has several limitations. Foremost, the study was single-center, retrospective in nature and involved a small group of patients. In addition, the study population was not homogeneous regarding the indications for ECLS (hemodynamic instability/cardiac arrest); however, all consecutive patients were analysed. According to the analysed data, cardiac arrest occurred in seven cases (19.4%) before the victim was identified. Among the patients with unwitnessed cardiac arrest, three (42%) survived and four (58%) died. The uneven distribution of patients to the unwitnessed CA group may carry the potential risk of bias. Similarly, the observed differences between surviving and deceased patients can also result from the witnessed/unwitnessed status of cardiac arrest. The small number of these patients and the relatively high survival rate did not support the subdivision of the study population into three groups.

Detailed v-a ECMO therapy parameters, such as blood flow, fresh gas flow, and rewarming rate, were not recorded. Because of the differences in the initial body core temperature and temperature gradient in individual patients, the time interval between the T1–T2 points was not standardized, and the calculated value of lactate kinetics was not time adjusted.

As we have mentioned, high doses of catecholamines can significantly affect lactate concentrations and thus lactate kinetics. Unfortunately, due to the retrospective nature of our study, we cannot retrieve data determining the precise doses of catecholamines.

The results of our study allow early selection of patients at high risk of therapy failure. However, further studies are needed to identify alternative therapies in these patients, such as reassigning the rewarming method, modifying the rewarming rate, the early initiation of renal replacement therapy, or using cytokine-absorbing filters.

## 5. Conclusions

Our results indicate that a high lactate concentration is an independent factor for the poor prognosis of hypothermic patients undergoing ECLS rewarming. Reduced value of lactate kinetics at 4 h after reaching 30 °C is also a factor of poor prognosis. Moreover, high potassium levels, persistently low pH values, HCO_3_ concentration, and BE values significantly increase the risk of therapy failure despite the implemented treatment.

## Figures and Tables

**Figure 1 ijerph-18-09764-f001:**
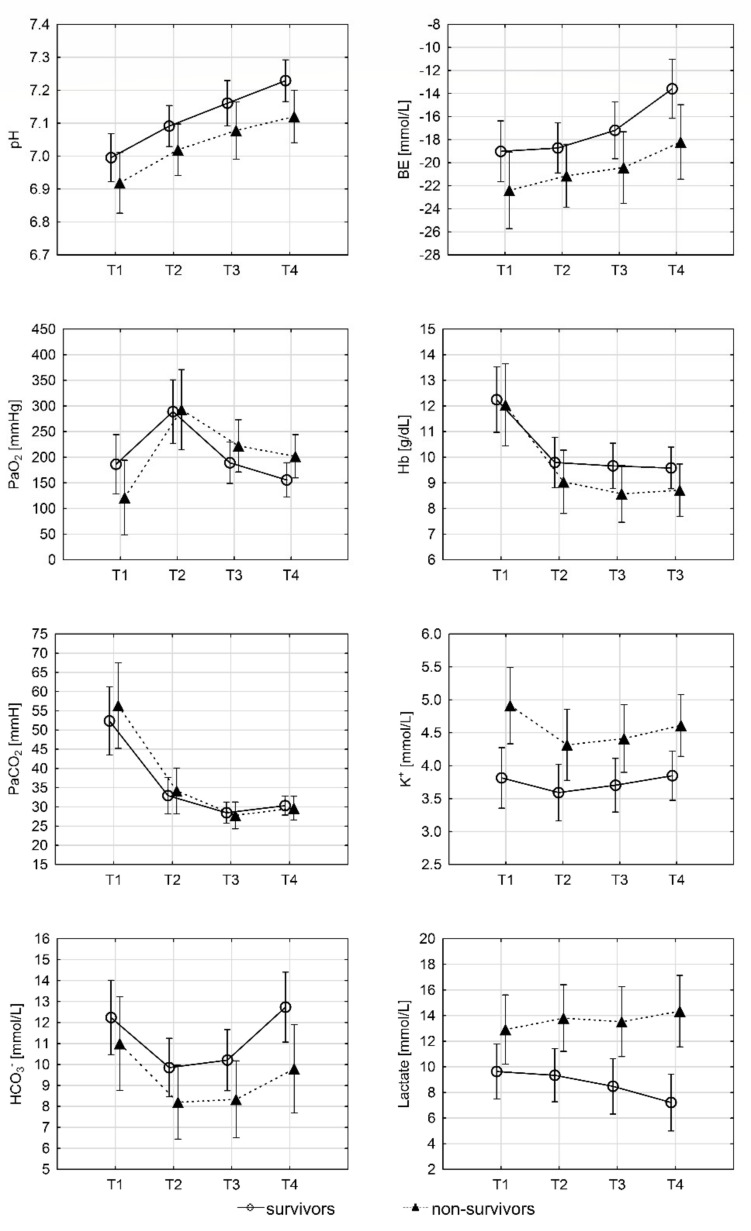
The mean values of blood pH, PaCO_2_, PaO_2_, HCO_3_, BE, and Hgb values are plotted at different time points. The bars represent a 95% confidence interval. T1—time at ECMO implantation, T2—time obtained Tc 30 °C, T3—2 h after Tc 30 °C, and T4—4 h after Tc 30 °C.

**Figure 2 ijerph-18-09764-f002:**
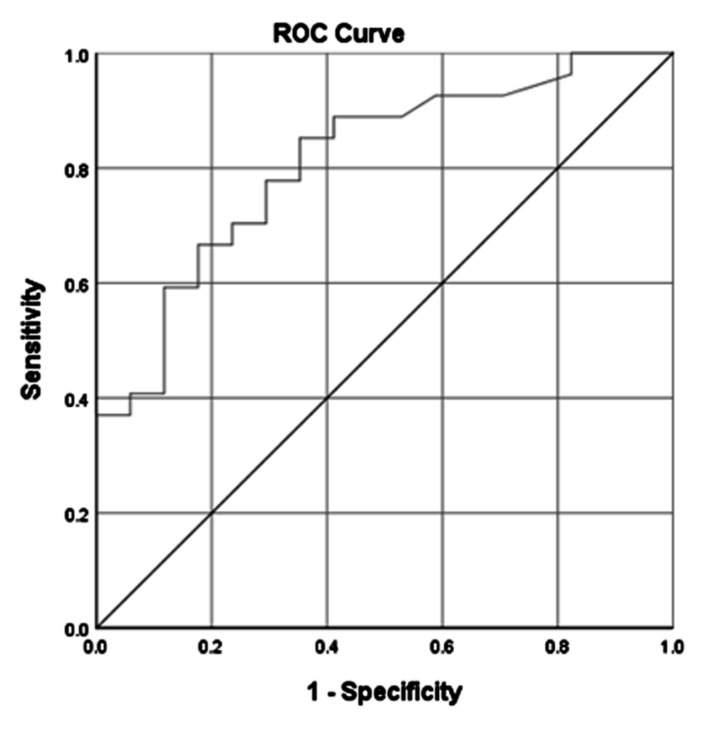
Receiver operating characteristic curve (ROC) analysis for the occurrence of death in relation to the lactate concentration at time point T4. AUROC (area under the receiver operating characteristic curve)—0.812. Standard error—0.065. *p* < 0.001. 95% CI—0.684–0.939. Cut-off point (95% CI) > 12.55. Sensitivity (%) of cut-off value—85. Specificity (%) of cut-off value—65.

**Table 1 ijerph-18-09764-t001:** Patients’ characteristics on admission.

Overall (*n* = 50)	Survival at ICU Discharge (*n* = 29)		Death (*n* = 21)		*p*
Age, Mean	55.30	±12.96	56.50	±16.87	0.783
Men, *n* (%)	24.00	82.8%	15.00	71.4%	0.336
pH	6.99	±0.17	6.89	±0.21	0.067
pCO_2_ (mmHg)	51.50	±20.03	56.50	±24.24	0.424
pO_2_ (mmHg)	92.00	(53.8–317.0)	73.70	(53.9–130)	0.265
HCO_3_ (mmol/L)	12.00	±4.37	10.50	±4.63	0.258
BE (mmol/L)	−19.30	±6.29	−23.20	±7.4	0.054
K+ (mmol/L)	3.90	±1.21	5.00	±1.61	0.008 *
Haemoglobin (g/dL)	12.30	±3.3	11.80	±3.07	0.566
Glucose (mmol/L)	8.10	(4.5–11.5)	6.30	(4.1–10.2)	0.520
Lactate (mmol/L)	9.80	±4.96	13.20	±5.73	0.028 *
Tc (°C)	23.90	±2.64	25.20	±2.41	0.090
CA	19	66.5%	17	81.0%	0.230
Time from CA Onset to v-a ECMO Implantation (min)	144.00	(120–195)	120.00	(67.0–240)	0.350

* *p* < 0.05.

**Table 2 ijerph-18-09764-t002:** Changes in lactate concentrations and kinetics in arterial blood gas from T1 to T4 time points.

Concentration	Survival		Death		*p*
T1 (*n* = 50)	9.80	±4.96	13.20	±5.73	0.028 *
T2 (*n* = 49)	9.40	±4.58	13.90	±5.94	0.004 *
T3 (*n* = 48)	8.50	±4.65	13.30	±6.50	0.005 *
T4 (*n* = 44)	7.20	±4.83	14.30	±6.90	<0.001 *
Difference in Concentration					
T2-T1 (*n* = 49)	−0.36	±2.06	0.73	±3.82	0.204
T3-T1 (*n* = 48)	−1.29	±3.46	0.44	±4.16	0.126
T4-T1 (*n* = 44)	−2.42	±4.49	1.44	±6.41	0.024 *
Lactate kinetics					
T2/T1 (*n* = 49)	0.78	(−11.3–15.5)	1.08	(−11.5–10.1)	0.490
T3/T1 (*n* = 48)	5.90	(−10.5–37.1)	1.70	(−26.1–21.1)	0.320
T4/T1 (*n* = 44)	21.20	(−14.1–56.8)	−2.60	(−16.0–13.5)	0.048 *
T3/T2 (*n* = 48)	0.00	(−8.8–33.1)	−1.77	(−14.3–19.7)	0.310
T4/T3 (*n* = 44)	24.00	(−4.5–40.0)	0.00	(−8.1–8.1)	0.008 *

* *p* < 0.05.

**Table 3 ijerph-18-09764-t003:** Distribution of the analysed clinical parameters in the survivors and those that died.

Variable	Survival (*n* = 29)		Death (*n* = 21)		*p*
NaHCO_3_ 8.4% Therapy 0–24 h (mL)	80	(40–80)	140	(100–200)	0.070
Transfusions of RCC (n) ≤4 units >4 units	18 11	(62) (38)	10 11	(48) (52)	0.310
Rewarming Rate (°C/h)	1.78	(1.35–2.90)	2.07	(1.55–2.60)	0.350
Duration of v-a ECMO Therapy (h)	23	(21–34)	9	(6–21)	<0.001 *
Length of ICU Hospitalisation (Days)	13	(8–22)	1	(1–3)	<0.001 *
Mechanical Ventilation (h)	164	(74–298)	29	(9–51)	<0.001 *
Diuresis 0–24 h (mL)	3300.00	(2400–4300)	650.00	(50–1050)	<0.001 *
Fluid therapy 0–24 h (mL)	10,889.70	(3338.56)	9431.00	(3881.64)	0.160

* *p* < 0.05.

**Table 4 ijerph-18-09764-t004:** Relationship between survival, age, gender, CA onset, body temperature, rewarming rate, and lactate concentration from T1 to T4 time points.

Variable	T1			T2			T3			T4		
	OR	95% CI	*p*	OR	95% CI	*p*	OR	95% CI	*p*	OR	95% CI	*p*
Age	0.966	(0.917–1.019)	0.205	0.954	(0.900–1.010)	0.104	0.955	(0.902–1.011)	0.115	0.965	(0.901–1.034)	0.315
Sex	0.573	(0.124–2.642)	0.475	0.384	(0.073–2.016)	0.258	0.272	(0.045–1.654)	0.157	0.043	(0.003–0.632)	0.022 *
CA (yes)	0.213	(0.036–1.267)	0.089	0.341	(0.057–2.061)	0.241	0.404	(0.063–2.611)	0.341	0.659	(0.079–5.469)	0.699
Tc	0.754	(0.557–1.019)	0.066	0.781	(0.569–1.072)	0.126	0.797	(0.569–1.116)	0.186	0.711	(0.450–1.126)	0.146
Rewarming Rate	0.728	(0.444–1.457)	0.472	0.728	(0.375–1.412)	0.347	0.790	(0.403–1.548)	0.492	0.400	(0.468–0.875)	0.107
Lactate concentration T1	0.851	(0.735–0.986)	0.032 *									
Lactate concentration T2				0.779	(0.649–0.934)	0.007 *						
Lactate concentration T3							0.779	(0.653–0.929)	0.005 *			
Lactate concentration T4										0.640	(0.468–0.875)	0.005 *

AUC	0.770	(0.630–0.899)		0.806	(0.685–0.926)		0.797	(0.671–0.922)		0.911	(0.829–0.993)	

* *p* < 0.05.

## Data Availability

The data that support the findings of this study are available on request from the corresponding author.

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
