# Peer review of "Prognosis of Hypothermic Patients Undergoing ECLS Rewarming—Do Alterations in Biochemical Parameters Matter?"

_ijerph, 2021, doi:10.3390/ijerph18189764_

Round 1
Reviewer 1 Report
The manuscript is generally well written, however, there are some points I would like to address:
Point 1: The authors use the term “lactate clearance”. A wide number of investigators have used this term to describe decreasing lactate levels, but, as pointed out by Vincent et al (The value of blood lactate kinetics in critically ill patients: a systematic review. Crit Care 20, 257; 2016), this is incorrect for two reasons: „The first is that the changes in lactate concentrations over time reflect changes in production and in elimination. The decrease in lactate over time may reflect decreased (over)production more than increased clearance by the liver and other organs. The specific study of lactate clearance would require intravenous injection of radiolabeled lactate, as has been done in several studies. The second reason why use of the term is incorrect is that “clearance” or “elimination” implies a progressive normalization of blood lactate concentrations, which is too simplistic. Blood lactate concentrations can have a complex evolution and may even increase over time…“
Please consider using the term “lactate kinetics” instead of lactate clearance.
Point 2: Please explain to the reader (in the methods section) why exactly you defined T2 when reaching 30°C. One could, for instance, ask why you did not start analysing the parameters in a 2-hours interval from the start of ECMO (i.e., T2 = 2 hours after T1)? And could you elucidate why you limited the analysis of the kinetics of the biochemical variables to just 4 hours after T2?
Point 3: Among the causes accidental hypothermia you mentioned drowning in 2 cases and avalanche in 1 case. Could you give some details regarding the cause of hypothermia in the remaining cases (also, for instance, if occurred indoors or outdoors…)?
Point 4: Table 1: Could you add the major comorbidities of the patients?
Point 5: In general, table and figure legends should be self-explaining by their own without the need to go back to the text to find information on the interpretation of the table/figure. For example: the parameters reported in table 1 are on admission. Yet, this is specified in the text only but should be written also in the legend of the table. Or in figure 1: are mean or median values reported? Do the bars represent the standard deviation?
Point 6: Table 2: Is the “lactate clearance” T3/T2 really 0,00? This does not seem to be the case, neither from the trend in figure 1 nor from the mean values at T2 and T3 reported in table 2. Please check this.
Point 7: Page 6, line 194: define abbreviation HT3
Point 8: Table 3: Bicarbonate: concentration? 8.4%?
Point 9: Did you use “cytokine removal filters” such as Cytosorb in some patients? If so, please specify in table 3
Point 10: You reported that catecholamines were needed for all patients. As you correctly pointed out in the discussion, catecholamines (in particular epinephrine) can increase lactate levels. Can you add e.g. the mean dose of epinephrine during ECMO to table 3?
Point 11: Figure 2: Please specify the AUROC?
Point 12: Could you (e.g. at the end of the discussion) tell if the results of your study could already be practice changing (and if so: how; and if not: give an outlook of further studies needed?)
Author Response
Dear Reviewer of International Journal of Environmental Research and Public Health,
Thank you very much for your patience and all the valuable comments and suggestions. I tried my best to address all your concerns. Please find my answers below.
Sincerely yours,
Dariusz Plicner on behalf of the authors.
- The authors use the term “lactate clearance”. A wide number of investigators have used this term to describe decreasing lactate levels, but, as pointed out by Vincent et al (The value of blood lactate kinetics in critically ill patients: a systematic review. Crit Care 20, 257; 2016), this is incorrect for two reasons: „The first is that the changes in lactate concentrations over time reflect changes in production and in elimination. The decrease in lactate over time may reflect decreased (over)production more than increased clearance by the liver and other organs. The specific study of lactate clearance would require intravenous injection of radiolabeled lactate, as has been done in several studies. The second reason why use of the term is incorrect is that “clearance” or “elimination” implies a progressive normalization of blood lactate concentrations, which is too simplistic. Blood lactate concentrations can have a complex evolution and may even increase over time…“
Please consider using the term “lactate kinetics” instead of lactate clearance.
Ad. 1 The phrase “lactate clearance” was changed to "the value of blood lactate kinetics", according to your suggestion.
The following section has been added to the Discussion text (lines 269-276):
“What should be noted, a significant number of publications describe lactate kinetics by the term clearance; however, as pointed out by Vincent et al., this term may be misleading [11]. Plasma lactate concentration depends on changes in production and elimination, while the term clearance refers only to the elimination of a substance from the plasma per unit time. It is therefore suggested that the term lactate kinetics be used instead of lactate clearance. In this regard, the existing publications can be analysed with a better understanding”.
- Please explain to the reader (in the methods section) why exactly you defined T2 when reaching 30°C. One could, for instance, ask why you did not start analysing the parameters in a 2-hours interval from the start of ECMO (i.e., T2 = 2 hours after T1)? And could you elucidate why you limited the analysis of the kinetics of the biochemical variables to just 4 hours after T2?
Ad. 2 Thank you for your valuable comment. We kindly propose to clarify the research strategy in the discussion. We have included the following paragraph in the Discussion text (lines 319-329) :
“In our study, we defined the time point T2 as the moment of reaching the core temperature of 30C, as it is considered the potential threshold of cardiovascular stabilization and the point for unaltered catecholamines action. According to the resuscitation guidelines, a core temperature of 30 is the moment of pharmacotherapy initiation, increasing the chances of successful defibrillation and return of spontaneous circulation [26,27,28]. These assumptions allowed us to standardize the study population in our study regarding the core temperature and the stage of resuscitation. We limited our study to 4 hours after the time point T2 because the number of deaths significantly reduced the group, which may have affected the statistical analysis”.
We also added the following references to the manuscript:
- Frei C, Darocha T, Debaty G, Dami F, Blancher M, Carron PN, et al. Clinical characteristics and outcomes of witnessed hypothermic cardiac arrest: A systematic review on rescue collapse. Resuscitation. 2019;137:41–8.
- Soar J, Nolan JP, Böttiger BW, Perkins GD, Lott C, Carli P, et al. European Resuscitation Council Guidelines for Resuscitation 2015. Resuscitation. 2015;95:100–47.
- Tveita T. Pharmacodynamics in hypothermia. Crit Care. 2012;16:A6, cc11264.
- Among the causes accidental hypothermia you mentioned drowning in 2 cases and avalanche in 1 case. Could you give some details regarding the cause of hypothermia in the remaining cases (also, for instance, if occurred indoors or outdoors…)?
Ad. 3 The following was added to the Results text (lines 158-160):
“The remaining patients suffered from exposure to cold air. However, specific data regarding these events and detailed patient characteristics are incomplete and not suitable for analysis”.
- Could you add the major comorbidities of the patients?
Ad. 4 As we mentioned above we do not have full patient’s characteristics, because specific group of patients. We added the following sentences to the Results section (lines 158-160):
“The remaining patients suffered from exposure to cold air. However, specific data regarding these events and detailed patient characteristics are incomplete and not suitable for analysis”.
- In general, table and figure legends should be self-explaining by their own without the need to go back to the text to find information on the interpretation of the table/figure. For example: the parameters reported in table 1 are on admission. Yet, this is specified in the text only but should be written also in the legend of the table. Or in figure 1: are mean or median values reported? Do the bars represent the standard deviation?
Ad. 5 Appropriate changes have been made to the Figures and Tables, according to your suggestions.
- Table 2: Is the “lactate clearance” T3/T2 really 0,00? This does not seem to be the case, neither from the trend in figure 1 nor from the mean values at T2 and T3 reported in table 2. Please check this.
Ad. 6 Thank you kindly for your comments. We have verified all the data reported in the manuscript. The values are correct. As per data distribution, concentration values are presented as means while the clearance values as medians. The term "clearance" has been replaced by "lactate kinetics".
- Page 6, line 194: define abbreviation HT3.
Ad. 7 We have made the amendments as suggested in the Results text (lines 205-207):
“The highest lactate concentration registered in a survivor was 21mmol/L, and which was measured at the T1 time point in a patient with preserved spontaneous circulation”.
- Table 3: Bicarbonate: concentration? 8.4%?
Ad. 8 We have made changes to Table 3, according to your suggestion.
- Did you use “cytokine removal filters” such as Cytosorb in some patients? If so, please specify in table 3.
Ad 9 Thank you for your question regarding the treatment provided. Cytosorb is not a routine form of therapy and no patient in our study received this form of treatment .
- You reported that catecholamines were needed for all patients. As you correctly pointed out in the discussion, catecholamines (in particular epinephrine) can increase lactate levels. Can you add e.g. the mean dose of epinephrine during ECMO to table 3?
Ad 10 The following was added to the Discussion text (lines 357-360):
“As we have mentioned, high doses of catecholamines can significantly affect lactate concentrations and thus lactate kinetics. Unfortunately, due to the retrospective nature of our study, we cannot retrieve data determining the precise doses of catecholamines”.
- Figure 2: Please specify the AUROC?
Ad. 11 We have made a change in the description of Figure 2, according to your suggestion:
- Could you (e.g. at the end of the discussion) tell if the results of your study could already be practice changing (and if so: how; and if not: give an outlook of further studies needed?)
Ad 12 Thank you for your suggestions. The following sentences has been added to the Discussion text (lines 361-365):
“The results of our study allow early selection of patients at high risk of therapy failure. However, further studies are needed to identify alternative therapies in these patients, such as reassigning the rewarming method, modifying the rewarming rate, early initiation of renal replacement therapy, or using cytokine absorbing filters”.
Reviewer 2 Report
Please see document attached.

Author Response
Dear Reviewer of International Journal of Environmental Research and Public Health,
Thank you very much for your patience and all the valuable comments and suggestions. I tried my best to address all your concerns. Please find my answers below.
Sincerely yours,
Dariusz Plicner on behalf of the authors.
- Did the rewarming speed differ among the groups? Could that have influenced the outcome?
Ad. 1 Thank you for your question. In the group of patients who survived, the rewarming rate was 1,78 °C/h and in the group of patients who died was 2,07 °C/h. The reported values did not differ significantly between the groups (p > 0.05).
We made a note about this in the Results text (lines 210-211):
“The rewarming rate ranged from 0.4-6 °C/h and did not differ significantly between groups”.
- In the survival group the average starting temperature is lower than in the deceased group. Assuming a roughly equal rewarming speed among the groups, that would indicate a systematically longer T1-T2 duration in the surviving group, which could bias the results. In any case, I would recommend indicating the rewarming speeds of both groups.
Ad. 2 Thank you very much for your valuable comment. To our opinion, with regard to literature data, rather body temperature and rewarming rate than duration of ECLS support may have an influence on survival. In our study population, rewarming rates did not differ significantly between groups.
- You state that your moderation analysis showed that there is no relationship between CA and survival. How about unwitnessed CA and survival? Could an uneven distribution of unwitnessed CA (i.e., most unwitnessed CA in the deceased group) skew your data? Could the observed differences among the surviving and the deceased patients stem from the witnessed / unwitnessed CA arrest status?
Ad. 3 Thank you kindly for the valuable comment. The following paragraph was added to the Discussion text (lines 341-350):
“According to the analyzed data, cardiac arrest occurred in 7 cases (19.4%) before the victim was identified. Among patients with unwitnessed cardiac arrest, 3 (42%) survived, and 4 (58%) died. The uneven distribution of patients to the unwitnessed CA group may carry the potential risk of bis. Similarly, the observed differences between surviving and deceased patients can also result from witnessed/unwitnessed status of cardiac arrest. The small number of these patients and the relatively high survival rate did not support the subdivision of the study population into three groups”.
- The phrase in line 171-173 is not quite clear to me. Do you want to say there was a significant difference between all the timepoints within the groups? I would doubt that. Or was there a significant difference among some timepoints in some parameters? In this case, I would recommend elaborating your data more clearly.
Ad. 4 The sentence has been rewritten for clarification, the Results section (lines 178-181):
“The blood pH, PaCO2, PaO2, HCO3, BE, and Hgb values changed significantly between subsequent time points in each group, although there were no significant differences in these values between the survivors and non-survivors groups (Figure 1)”.
- In table 2 you analyze the lactate concentration, difference in concentration, and clearance at various time points. However, at T4 there are 6 patients less than at T1. Would you get the same results if you just included the 44 patients that survived till T4? Could for example the increase in lactate concentration between T3 and T4 be a consequence of 4 people dying and therefore skewing the results?
Ad. 5 Thank you very much for your valuable comment. The aim of this study was to analyse changes in the acid-base balance parameters and lactate kinetics to indicate predictors of clinical outcome. We think that exclusion of patients who died shortly after could distort the results. However, we share your opinion that including these subjects to the study could also misinterpret the findings.
- How does the use of vasopressor compare between the group? You state that vasopressors were used in all the patients. Could a higher lactate concentration be connected to a more extensive use of vasopressors?
Ad. 6 Thank you for your valuable comment. The following was added to the Discussion (lines 357-360):
“As we have mentioned, high doses of catecholamines can significantly affect lactate concentrations and thus lactate kinetics. Unfortunately, due to the retrospective nature of our study, we cannot retrieve data determining the precise doses of catecholamines”.
Minor comments:
- Why did you plot the lactate concentration for each time point in a different row in table 4?
Ad. 1 Thank you for your question. We hoped that this way of presenting the results is the most efficient way.
Round 2
Reviewer 2 Report
Congratulations on your work.